

# Similarities and differences: species and diet impact gut microbiota of captive pheasants

Yushuo Zhang, Xin He, Xiuhong Mo, Hong Wu and Dapeng Zhao

Tianjin Key Laboratory of Conservation and Utilization of Animal Diversity, College of Life Sciences, Tianjin Normal University, Tianjin, China

## ABSTRACT

The fecal microbiota plays an important role in maintaining animal health and is closely related to host life activities. In recent years, there have been an increasing number of studies on the fecal microbiota from birds. An exploration of the effects of species and living environments on the composition of gut microbiota will provide better protection for wildlife. In this study, non-injury sampling and 16S rDNA high-throughput sequencing were used to investigate the bacterial composition and diversity of the fecal microbiota in silver pheasants (*Lophura nycthemera*) and golden pheasants (*Chrysolophus pictus*) from Tianjin Zoo and Beijing Wildlife Park. The results showed that the abundance of Firmicutes was the highest in all fecal samples. At the genus level, *Bacteroides* was the common dominant bacteria, while there were some differences in other dominant bacteria genera. There were significant differences in fecal microbial composition between the golden pheasants from Tianjin Zoo and Beijing Wildlife Park. The metabolic analysis and functional prediction suggested that the gut microbiota composition and host metabolism were influenced by dietary interventions and living conditions. The results of this study provide the basis for further research of intestinal microbial of *L. nycthemera* and *C. pictus*, and valuable insights for conservation of related species.

## INTRODUCTION

The gut microbiota of animals contains complex microbes which can regulate host digestion (*Clemente et al., 2012*), metabolism (*Valdes et al., 2018*), and immune responses (*Waite & Taylor, 2014*). Previous studies have shown that gut microbial communities can reflect phylogenetic relationships and can further help us understand animal health (*Palinauskas et al., 2022*; *Viney, 2019*). Unlike other vertebrates, birds have short gastrointestinal tracts and short food retention time to support the requirement of flight (*Kohl, 2012*). Avian gut microbes are mainly composed of Firmicutes, Proteobacteria, Actinobacteria, and Bacteroidota (*Waite & Taylor, 2014*). Avian gut microbial communities are affected by the host species and living conditions, including environmental factors and diet (*Bodawatta et al., 2022*; *Kohl, 2012*; *Sun et al., 2022*). For instance, *Yang, Deng & Cao (2016)* found that the common dominant phyla included Firmicutes, Proteobacteria, and Actinobacteria among three wild goose species (white-

Corresponding authors
Hong Wu, skywuhong@tjnu.edu.cn
Dapeng Zhao, skyzdp@tjnu.edu.cn

fronted geese, bean geese, and swan geese), but the proportion of these common dominant phyla was different among species. The finding suggests that the host species is the potential driver leading the differentiation of goose gut microbiota. *Laviad-Shitrit et al. (2019)* found that there was a correlation between species phylogeny and gut microbial communities among four wild waterbird species (great cormorants, little egrets, black-crowned night herons and black-headed gulls). *Mohsin Bukhari et al. (2022)* found that, under captive conditions in Avian Conservation and Research Center, the gut microbiota of ring-necked pheasants was dominated by Firmicutes, Actinobacteriota, and Proteobacteria, with *Bacillus*, *Oceanobacillus*, and *Teribacillusas* as the dominant genera, whereas the gut microbiota of green pheasants was dominated by Firmicutes, Proteobacteria, and Bacteroidota, with *Bacillus* and *Lactobacillus* as the dominant genera. Thus, even under the same living environment, including the temperature, humidity and other conditions, as well as the type of food provided by the external environment, the host species is considered to be the main factor leading to the different composition and characteristics of avian gut microbiota.

The gut microbiome of birds was derived primarily from the environment since birth. Environmental factors affect the behavior, foraging, and growth of birds, thus are important in shaping the composition and characteristics of avian gut microbes (*Liu et al., 2022*; *Xie et al., 2016*; *Yao et al., 2023*). *Chi et al. (2019)* found that, for both wild and captive bharals, Firmicutes and Bacteroidetes were the common dominant phyla while *Bacteroides* and *Alistipes* were the common dominant genera. The researchers also found that the abundance of Firmicutes in wild bharals was significantly higher than that in captive bharals whereas the abundance of Bacteroidetes in captive bharals were significantly higher than that in wild bharals. *Wang et al. (2020)* found that, although there were four common abundant phyla (Firmicutes, Proteobacteria, Actinobacteria, and Bacteroidota) of gut microbiota in wild black-necked cranes living in six overwintering areas, the variance in alpha and beta diversities were found among different living areas. Similarly, *Gu & Zhou (2021)* found the similar discrepancy in both community composition and alpha-diversity of gut microbiota in wild hooded cranes living at Poyang Lake, Shengjin Lake, and Caizi Lake respectively.

Silver pheasants (*Lophura nycthemera*) and golden pheasants (*Chrysolophus pictus*) are classified under the Phasianidae family within the order Galliformes, but belong to two different genera, *Lophura* and *Chrysolophus*, respectively. Silver pheasants are mainly distributed in China, Cambodia, Myanmar, Thailand, and Vietnam (*Dong et al., 2013*), while the golden pheasant is an endemic species in China (*Liu et al., 2021*). Many studies have focused on the activity rhythms, habitat selection and captive management of these two pheasant species recently, for example, *Kullu et al. (2016)* studied the effect of nitrogen and mineral retention by increasing the dietary supply of carotenoids in captive golden pheasants, as well as the influence of different levels of green vegetables on egg production performance (*Kullu et al., 2017*). However, the gut microbiome of *L. nycthemera* and *C. pictus* is still poorly known. For instance, *Mushtaq et al. (2021)* found that *Escherichia coli* is predominant isolated from fecal samples for both *L. nycthemera* and *C. pictus* in captive conditions.

Considering that species and habitat environment mainly work as potential drivers of diversity in avian gut microbiota (*Wang et al., 2022*), this study is the first time to investigate differences in the composition of gut microbiota in *L. nycthemera* and *C. pictus* under different captive environments based on 16S ribosomal DNA (rDNA) high-throughput sequencing technology. By investigating the relationship between fecal microbiota composition and living conditions for these two species of pheasants, it will provide scientific reference for the *ex-situ* conservation of pheasants in captivity.

## MATERIALS AND METHODS

### Sample collection

Twelve fresh fecal samples for each pheasant species were collected from May 2020 to May 2021, based on the non-invasive sampling technique (*de Flamingh et al., 2023*). All fecal samples were divided into four groups, namely SCB (six silver pheasants from Beijing), SCT (six silver pheasants from Tianjin), GCB (six golden pheasants from Beijing) and GCT (six golden pheasants from Tianjin) (Table 1). The sample collection complied with the current laws of China and were approved by Animal Ethics Committee of Tianjin Normal University. Fecal samples of *L. nycthemera* and *C. pictus* were collected without direct contact with the animals. We collected fecal samples immediately after animals had defecated in their cages, and stored samples in a portable ice box. Samples were then transported and stored at −80 °C.

### DNA extraction and 16S rDNA sequencing

Total DNA was extracted from fecal samples using the CTAB method (*Arseneau, Steeves & Laflamme, 2017*). DNA quality was assessed by electrophoresis on a 1% (w/v) agarose gel and purity was determined on a NanoDrop 2000 UV-vis spectrophotometer. Following successful DNA extraction, the V3–V4 region of the 16S rDNA gene was amplified using the following specific PCR primers: Forward primer 341F (5′-CCTACGGGNGGCWGCAG-3′) and reverse primer 805R (5′-GACTACHVGGGTATCTAATCC-3′). The PCR reaction (total volume of 20 μL) included template DNA, primers, DNA polymerase, 5 × Fast Pfu Buffer, and dd $H_2O$. PCR amplification products were assessed by electrophoresis on a 2% (w/v) agarose gel. Finally, the purified amplicons were analyzed on an Illumina MiSeq platform (Illumina, San Diego, CA, USA), according to the standard protocols by Majorbio Bio-Pharm Technology Co. Ltd. (Shanghai, China). The libraries were constructed by double-ended sequencing and didn't spiked with phiX libraries. The raw reads were deposited into the NCBI Sequence Read Archive (SRA) database under PRJNA941118 (accession number: SRR24436453–SRR24436476).

### Bioinformatics and statistical analysis

The microbial communities of fecal samples from *L. nycthemera* and *C. pictus* were studied, and the data obtained were quality filtered using QIIME (version 1.9.1) (*Caporaso et al., 2010*) after 16S rDNA high-throughput sequencing (*D'Amore et al., 2016*; *Jiang & Takacs-Vesbach, 2017*; *Jiang et al., 2022*). The sequencing results consisted of double-ended sequence data. Initially, the pairs of reads were merged into a single

**Table 1 Information regarding the silver pheasants and golden pheasants used in this study.**

| Group | Species | Living environment | Number | Diet |
|-------|---------|--------------------|--------|------|
| SCT | Silver pheasant | Tianjin Zoo | 6 | Feed pelleted feed and chopped vegetables once a day. |
| GCT | Golden pheasant | Tianjin Zoo | 6 | Feed pelleted feed and chopped vegetables once a day. |
| SCB | Silver pheasant | Beijing Wildlife Park | 6 | Feed pheasant feed, vegetables and fruits once a day. |
| GCB | Golden pheasant | Beijing Wildlife Park | 6 | Feed pheasant feed, vegetables and fruits once a day. |

sequence based on the overlap between PE reads. Subsequently, quality control and filtering of sequencing data was conducted to remove low-quality sequences and chimera sequences. High-quality sequences were then compared, clustered, and classified to obtain information on the composition and diversity of microbial communities. Operational taxonomic unit (OTU) sequences with 97% similarity were annotated and analyzed using RDP classifier (version 2.11) according to the silva138/16s_bacteria database. Venn diagrams were generated to compare the numbers of shared OTUs and unique OTUs among the fecal microbial communities from different groups using R (version 3.3.1). The microbial composition of each fecal sample from the phylum to genus level were presented by community bar-plots. Alpha diversity indices including Chao, Ace, Shannon and Simpson indices were calculated using Mothur (version 1.30.2) to reflect the abundance and diversity of microbial communities (*Schloss et al., 2009*). The coverage indices calculated using Mothur reflect whether the sequencing results represent the real situation of the microorganisms in each sample. A Wilcoxon rank test was used for comparisons between different groups and $p \leq 0.05$ was considered statistically significant by using the FDR method. Principal Co-ordinates Analysis (PCoA) based on weighted and unweighted UniFrac distances were carried out to determine differences between two groups specifically (groups based on different species, different environments), and an Analysis of Similarities (ANOSIM) test based on an R vegan package was used to compare the variability between different groups (*Van Horn et al., 2016*).

## Predicted the function of gut microbiota by PICRUSt

PICRUSt is a software package that predicts the functional capabilities of microbial communities. In this study, PICRUSt was used to predict the potential functions of each fecal sample based on 16S rRNA gene sequencing data (*Langille et al., 2013*). The genes, their function and the abundance of metabolic pathways were predicted and summarized into the Kyoto Encyclopedia of Genes and Genomes (KEGG) database which is a systematic analysis of gene function, and genome information. By comparing the gut microbial data of *L. nycthemera* and *C. pictus* to the database of orthologous groups Cluster of Orthologous Groups of proteins (COG) and KEGG (*Kanehisa, 2019*; *Kanehisa et al., 2023*; *Kanehisa & Goto, 2000*; *Tatusov et al., 2000*), we obtained the corresponding functional and metabolic pathway prediction information for phenotypic prediction using BugBase (*Ward et al., 2017*). R software (v4.1.2; *R Core Team, 2021*) was utilized for statistical analyses and visualization of the identified pathways.
## RESULTS

### Analysis of gut microbiota composition from different groups

A total of 24 fecal samples from both species were analyzed and 1,700,701 optimized sequences were obtained, with an average length of 411 bp (Table S1). According to 97% similarity, 2,949 OTUs were obtained and could be classified into 42 phyla, 130 classes, 279 orders, 428 families, and 847 genera. Of the 2,949 OTUs, 745 OTUs were shared in all four groups, whereas 183, 169, 122, and 756 OTUs were unique to group SCT, SCB, GCT, and GCB, respectively (Fig. 1A). At the genus level, there were 258 genera shared by four groups, while the group GCB contained the largest unique genus (Fig. 1B).

At the phylum level, Firmicutes was dominant in all four groups (SCT: 47.56%; SCB: 43.91%; GCT: 72.48%; GCB: 35.22%), followed by Bacteroidota (SCT: 21.89%; SCB: 26.56%; GCT: 15.74%; GCB: 14.52%); Actinobacteriota (SCT: 14.35%; SCB: 9.50%; GCT: 6.50%; GCB: 20.17%) and Proteobacteria (SCT: 10.84%; SCB: 13.88%; GCT: 2.58%; GCB: 18.79%) (Fig. 2A, Table 2).

At the genus level, the dominant genera shared by SCT and SCB included *Bacteroides* (SCT: 11.07%, SCB: 13.24%), *Burkholderia-Caballeronia-Paraburkholderia* (SCT: 4.50%, SCB: 10.38%), and *norank _f__norank_o__Clostridia_UCG-014* (SCT: 3.36%, SCB: 3.76%). The remaining dominant genera in the SCT were *Streptococcus* (5.24%), *Bifidobacterium* (5.06%), *Romboutsia* (4.43%), *Clostridium_sensu_stricto_1* (3.87%), and *Collinsella* (3.35%); the remaining dominant genera of SCB were *unclassified_f__Lachnospiraceae* (6.97%), *Rikenellaceae_RC9_gut_group* (4.66%), *Ruminococcus_torques_group* (4.11%), *Olsenella* (3.50%), and *Desulfovibrio* (3.48%) (Fig. 2B, Table 3). The common genera in both GCT and GCB included *Bacteroides* (GCT: 6.63%, GCB: 4.47%), *Subdoligranulum* (GCT: 3.47%, GCB: 3.46%), and *Ruminococcus_torques_group* (GCT: 2.72%, GCB: 6.29%). The remaining dominant genera in GCT were *Clostridium_sensu_stricto_1* (15.22%), *Lactobacillus* (12.18%), *Anaerosporobacter* (9.10%), *unclassified_f__Lachnospiraceae* (3.01%), *Faecalibacterium* (2.93%); and the remaining dominant genera in GCB were *Burkholderia-Caballeronia-Paraburkholderia* (14.07%), *Olsenella* (7.54%), *Bifidobacterium* (4.24%), *Streptococcus* (4.06%), and *Rikenellaceae_RC9_gut_group* (3.99%) (Fig. 2B, Table 3).

The microbial composition from all 24 fecal samples divided into four groups in this study was also compared at the class, order, and family levels (Figs. S1 and S2).

### Analysis of differences in gut microbiota between different groups

Based on Wilcoxon rank test, the top abundance at the phylum level were compared and the results showed that the average relative abundance of Firmicutes in GCT (72.48%) was significantly higher than that in GCB (35.22%), and the average relative abundance of Patescibacteria, Chloroflexi, Nitrospirota, Verrucomicrobiota, and Methylomirabilota in GCT were significantly lower than that in GCB. The main difference between GCB and GCT groups was the living condition. Similarly, the gut microbiota of *L. nycthemera* in different living conditions also showed significant differences, such as the abundance of Fusobacteriota in SCT was significantly higher than that in SCB, while, the average relative

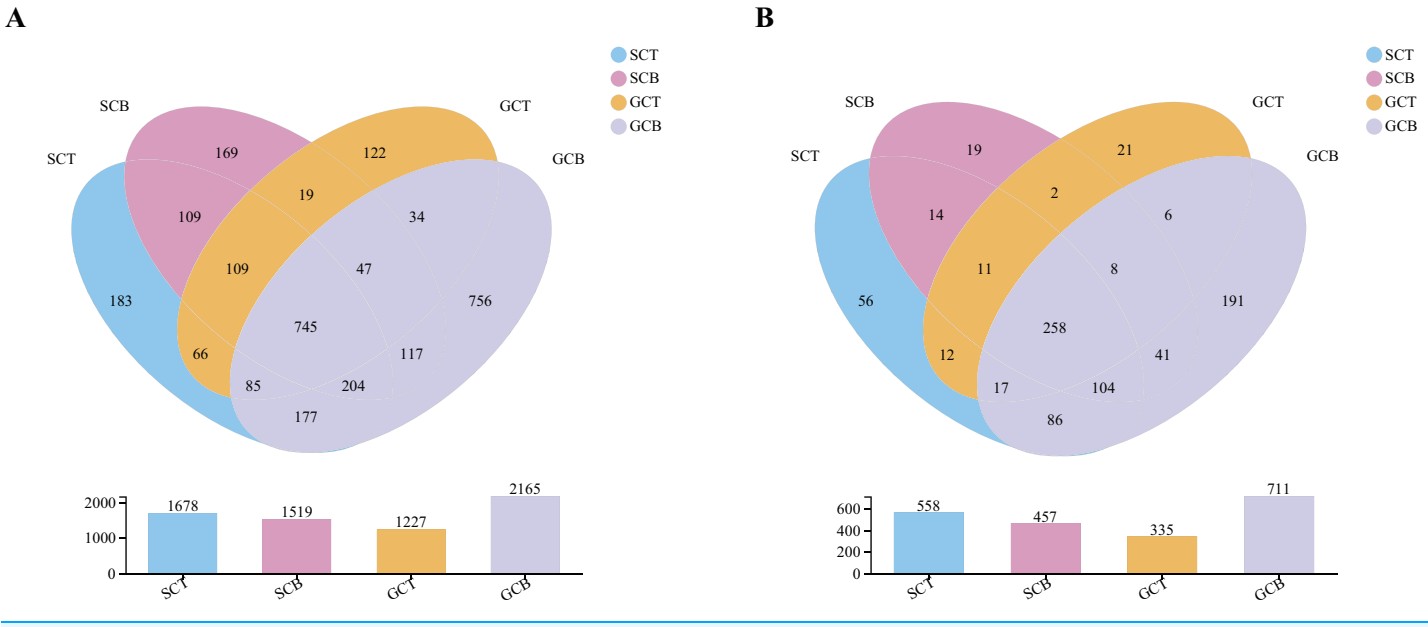

**Figure 1 Venn diagrams analysis of microbiota at levels of OTU (A)/genus (B).**

abundance of Verrucomicrobiota in SCT was significantly lower than that in SCB. In addition, the gut microbiota of different species varies significantly even when they live in the same environment. For example, the average relative abundance of Fusobacteriota, Chloroflexi, Nitrospirota in GCT were significantly lower than that in SCT, the average relative abundance of Deferribacterota in SCB was significantly higher than that in GCB (Fig. 3).

Wilcoxon rank test analysis of the top genera revealed that the abundance of *Bifidobacterium*, *Megamonas*, and *Solobacterium* in SCT were significantly higher than that in SCB, and the abundance of *Clostridium_sensu_stricto_1*, *Anaerosporobacter*, and *Cellulosilyticum* in GCT were significantly higher than that in GCB. The main difference between GCB/GCT and SCB/SCT groups was the living condition. Similarly, there were significant interspecific differences under the same living condition. For example, the abundance of *Christensenellaceae_R-7_group*, *UCG-005*, and *norank_f__norank_o__ Clostridia_vadinBB60_group* in SCB were significantly higher than that in GCB. Further, the abundance of *Subdoligranulum*, *Bifidobacterium*, and *norank_f__norank_o__ Saccharimonadales* in SCB were significantly lower than that in GCB.

The abundance of *Psychrobacter*, *Fusobacterium*, *UCG-002* in SCT were significantly higher than in GCT, while, the abundance of *Cellulosilyticum* in SCT was significantly lower than that in GCT (Fig. 4).

## Differences in alpha and beta diversity among four groups

The curve trends for all samples were similar, thus four groups had similar abundance and uniformity in terms of gut microbiota (Fig. S3). The alpha diversity index including Chao's index, Ace's index, Simpson's index, and Shannon's index were calculated (Table S2).

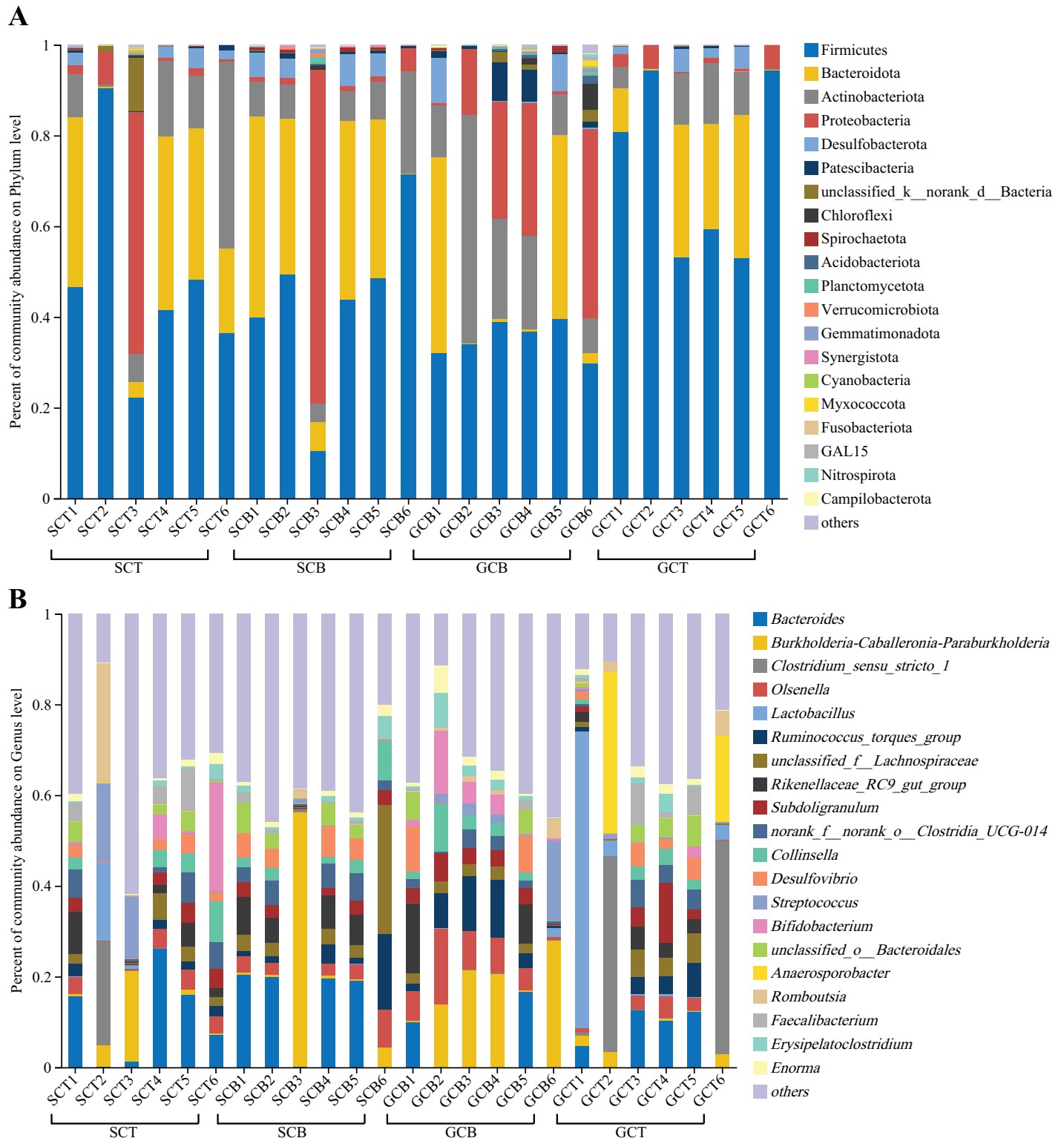

**Figure 2 Microbial composition of all fecal samples at the phylum/genus level.** (A) Bar plots showing the top 20 phyla in terms of relative abundance in all samples; (B) bar plots showing the top 20 genera in terms of relative abundance in all samples. A relative abundance of less than 1% and no annotation results were classified as "others."

**Table 2 Mean relative abundance of the 10 most abundant taxa at the phylum level.**

| Sample group | Top 10 abundant phyla (%) |
| --- | --- |
| SCT | Firmicutes (47.56) |
| | Bacteroidota (21.89) |
| | Actinobacteriota (14.35) |
| | Proteobacteria (10.84) |
| | unclassified_k__norank_d__Bacteria (2.22) |
| | Desulfobacterota (2.00) |
| | Patescibacteria (0.31) |
| | Fusobacteriota (0.15) |
| | Cyanobacteria (0.12) |
| | Chloroflexi (0.09) |
| SCB | Firmicutes (43.91) |
| | Bacteroidota (26.56) |
| | Proteobacteria (13.88) |
| | Actinobacteriota (9.50) |
| | Desulfobacterota (3.67) |
| | Patescibacteria (0.52) |
| | Spirochaetota (0.44) |
| | Verrucomicrobiota (0.28) |
| | Synergistota (0.24) |
| | Planctomycetota (0.22) |
| GCT | Firmicutes (72.48) |
| | Bacteroidota (15.74) |
| | Actinobacteriota (6.50) |
| | Proteobacteria (2.58) |
| | Desulfobacterota (2.30) |
| | Patescibacteria (0.16) |
| | unclassified_k__norank_d__Bacteria (0.07) |
| | Synergistota (0.05) |
| | Spirochaetota (0.03) |
| | Campilobacterota (0.02) |
| GCB | Firmicutes (35.22) |
| | Actinobacteriota (20.17) |
| | Proteobacteria (18.79) |
| | Bacteroidota (14.52) |
| | Patescibacteria (3.23) |
| | Desulfobacterota (3.13) |
| | Chloroflexi (1.22) |
| | unclassified_k__norank_d__Bacteria (1.07) |
| | Acidobacteriota (0.49) |
| | Spirochaetota (0.33) |

**Table 3  Mean relative abundance of the 10 most abundant taxa at the genus level.**

| Sample group | Top 10 abundant genera (%) |
| --- | --- |
| SCT | *Bacteroides* (11.07) |
| | *Streptococcus* (5.24) |
| | *Bifidobacterium* (5.06) |
| | *Burkholderia-Caballeronia-Paraburkholderia* (4.50) |
| | *Romboutsia* (4.43) |
| | *Clostridium_sensu_stricto_1* (3.87) |
| | *norank_f__norank_o__Clostridia_UCG-014* (3.36) |
| | *Collinsella* (3.35) |
| | *Rikenellaceae_RC9_gut_group* (3.06) |
| | *Faecalibacterium* (3.04) |
| SCB | *Bacteroides* (13.24) |
| | *Burkholderia-Caballeronia-Paraburkholderia* (10.38) |
| | *unclassified_f__Lachnospiraceae* (6.97) |
| | *Rikenellaceae_RC9_gut_group* (4.66) |
| | *Ruminococcus_torques_group* (4.11) |
| | *norank_f__norank_o__Clostridia_UCG-014* (3.76) |
| | *Olsenella* (3.50) |
| | *Desulfovibrio* (3.48) |
| | *Collinsella* (3.09) |
| | *unclassified_o__Bacteroidales* (2.97) |
| GCT | *Clostridium_sensu_stricto_1* (15.22) |
| | *Lactobacillus* (12.18) |
| | *Anaerosporobacter* (9.10) |
| | *Bacteroides* (6.63) |
| | *Subdoligranulum* (3.47) |
| | *unclassified_f__Lachnospiraceae* (3.01) |
| | *Faecalibacterium* (2.93) |
| | *Ruminococcus_torques_group* (2.72) |
| | *unclassified_o__Bacteroidales* (2.64) |
| | *norank_f__norank_o__Clostridia_UCG-014* (2.50) |
| GCB | *Burkholderia-Caballeronia-Paraburkholderia* (14.07) |
| | *Olsenella* (7.54) |
| | *Ruminococcus_torques_group* (6.29) |
| | *Bacteroides* (4.47) |
| | *Bifidobacterium* (4.24) |
| | *Streptococcus* (4.06) |
| | *Rikenellaceae_RC9_gut_group* (3.99) |
| | *Subdoligranulum* (3.46) |
| | *Collinsella* (3.36) |
| | *Desulfovibrio* (3.02) |

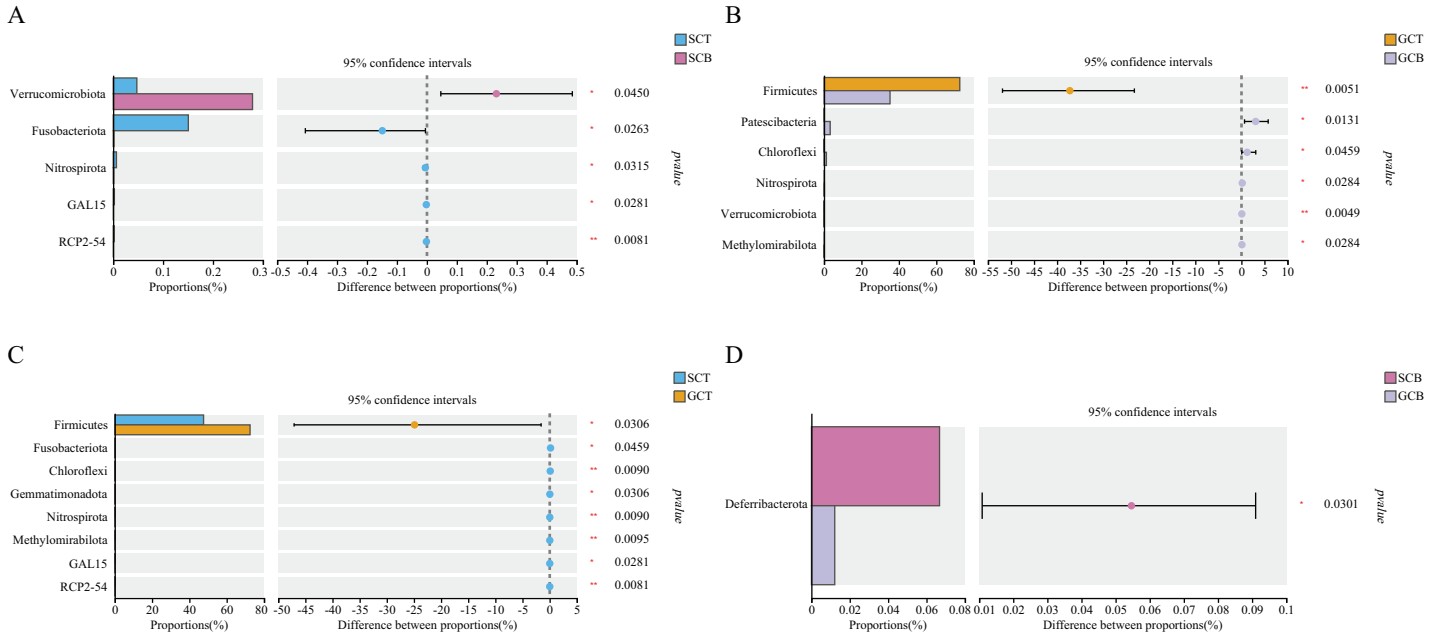

**Figure 3 Comparison of relative abundance of gut microbiota among different groups at the phylum level.** The significant differences in relative abundance between SCT and SCB (A), GCT and GCB (B), SCT and GCT (C), SCB and GCB (D) based on Wilcoxon rank test by using the FDR method. (*0.01 < P < 0.05, **0.001 < P < 0.01).

Under the same living condition in Tianjin Zoo, both ACE and Chao indices in SCT (824.4 and 839, respectively) were higher than that in GCT (659.9 and 633, respectively), while the Simpson index in SCT (0.06) was lower than that in GCT (0.13). Similar results were observed when comparison between SCB and GCB was conducted. The comparison of diversity differences of the same species in different environments showed that the ACE (SCT = 824.4, SCB = 870.3), Chao1 (SCT = 839, SCB = 867.1), Simpson (SCT = 0.06, SCB = 0.08), and Shannon (SCT = 4.11, SCB = 4.34) indices were not significantly different. The same condition appeared in the index's comparison between GCT and GCB. Thus, the results revealed that there were no significant differences in the diversity and abundance of gut microbiota between four groups (Fig. S4). However, Coverage indices of four groups were above 99.7%, indicating these data could adequately reflect the true situation of microorganisms in fecal samples for both *L. nycthemera* and *C. pictus*.

Principal Co-ordinates Analysis (PCoA) was used to evaluate the beta diversity of fecal microbial composition (Fig. 5). Based on the weighted Unifrac distances, the contribution rates of PC1 and PC2 were 46.32% and 28.5%, respectively. The contribution rates of PC1 and PC2 were 38.77% and 17.73%, respectively based on the unweighted Unifrac distances (Fig. 5). The results showed that there was a significant difference between GCT and GCB under both weighted_unifrac (R = 0.4222, P = 0.0050) and unweighted_unifrac (R = 0.1796, P = 0.0910), leading the complete separation between these two groups. However, there was no significant difference in beta diversity among individuals of *L. nycthemera* from different living conditions (P > 0.05, Fig. S5).

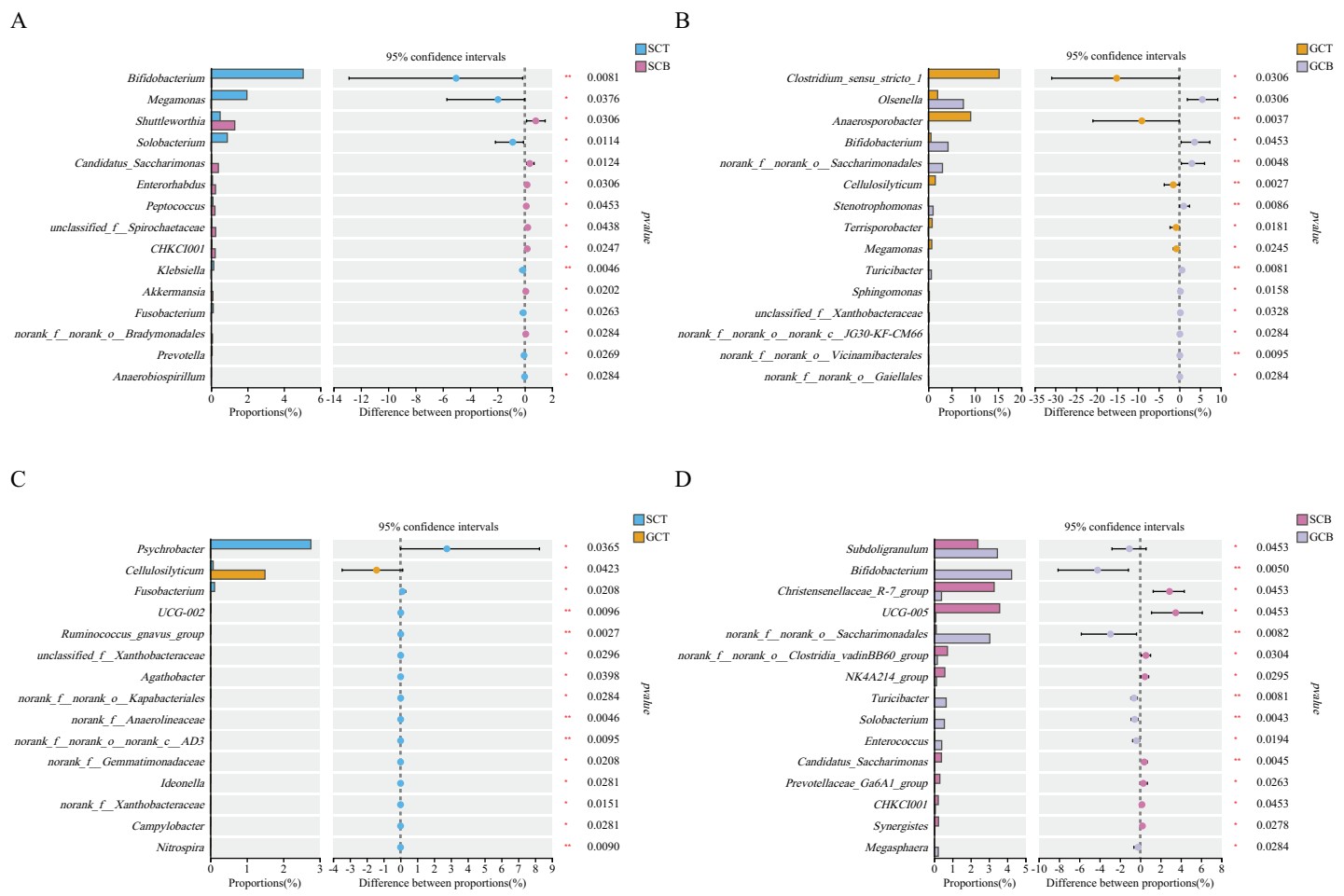

**Figure 4 Comparison of relative abundance of gut microbiota among different groups at the genus level.** The significant differences in relative abundance between SCT and SCB (A), GCT and GCB (B), SCT and GCT (C), SCB and GCB (D) based on Wilcoxon rank test by using the FDR method. (*0.01 < $P$ < 0.05, **0.001 < $P$ < 0.01).

The comparisons of gut microbiome profile were performed by linear discriminant analysis (LDA) effect size (LEfSe) in order to examine differences among the four groups (Fig. S6).

## Gut microbiota functional profile prediction

Based on the 16S rDNA sequencing results, the functional composition of COG was relatively similar in all samples, mainly related to the processing of genetic information such as transcription, translation, replication, transport, and metabolism of substances, as well as various metabolic pathways related to life activities (Fig. S7).

Statistical analysis on the abundance of KEGG metabolic pathways at Level 1 revealed that all four groups had the highest abundance in the metabolism pathway, with the higher relative abundance in Level 2 categories such as global and overview maps, carbohydrate metabolism, and amino acid metabolism (Fig. 6). The relative abundance of carbohydrate metabolism, amino acid metabolism, membrane transport and metabolism of vitamins were compared to explore impacts caused by environment and species. Among them,

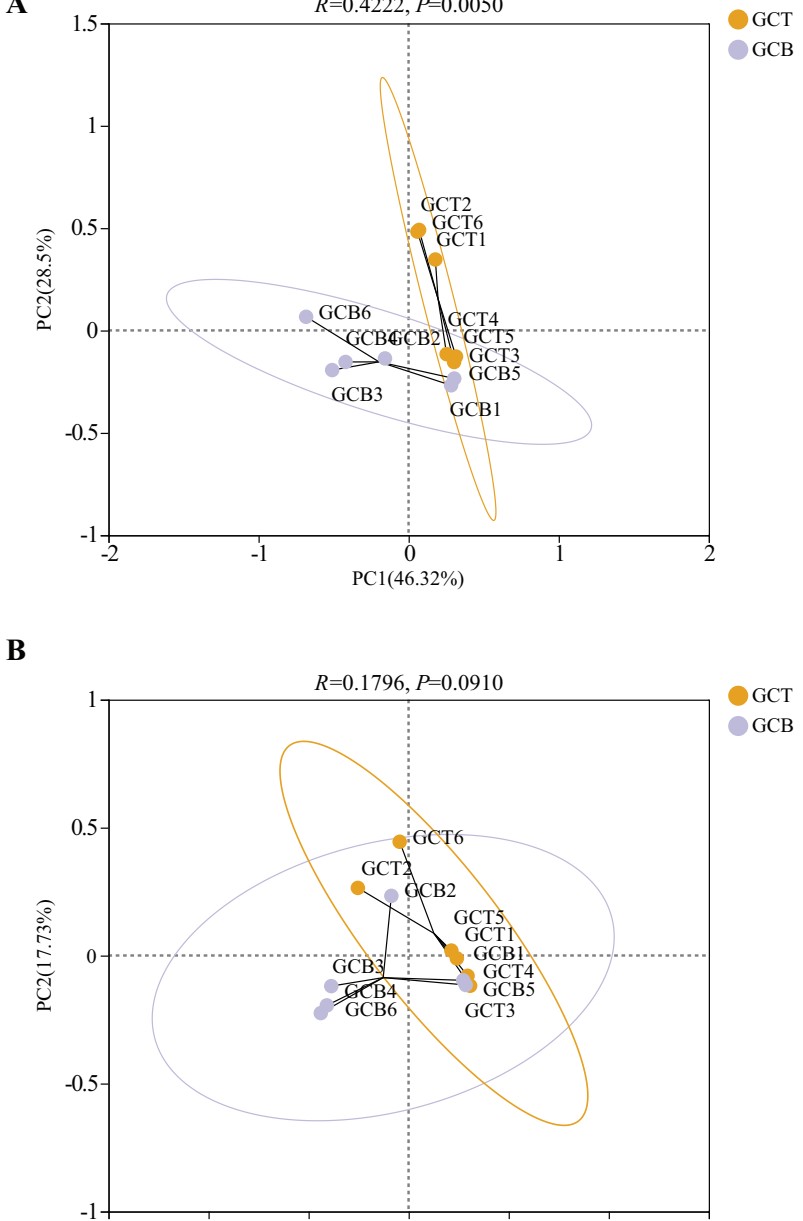

**Figure 5** PCoA analysis of the difference between GCT and GCB based on the weighted Unifrac distances (A) and unweighted Unifrac distances (B).

carbohydrate and amino acid metabolism pathways showed differences between species under different living conditions (GCB *VS* GCT) ($P < 0.05$) (Figs. S8A, S8B). Using BugBase phenotype prediction analysis, seven phenotypes of gut microbiota in all fecal samples were predicted. The relative abundance of Gram-negative and biofilm forming micro-organisms were significantly higher in GCB than that in GCT, while the relative abundance of Gram-positive bacteria was significantly lower in GCB than that in GCT (Figs. S8C–S8E).

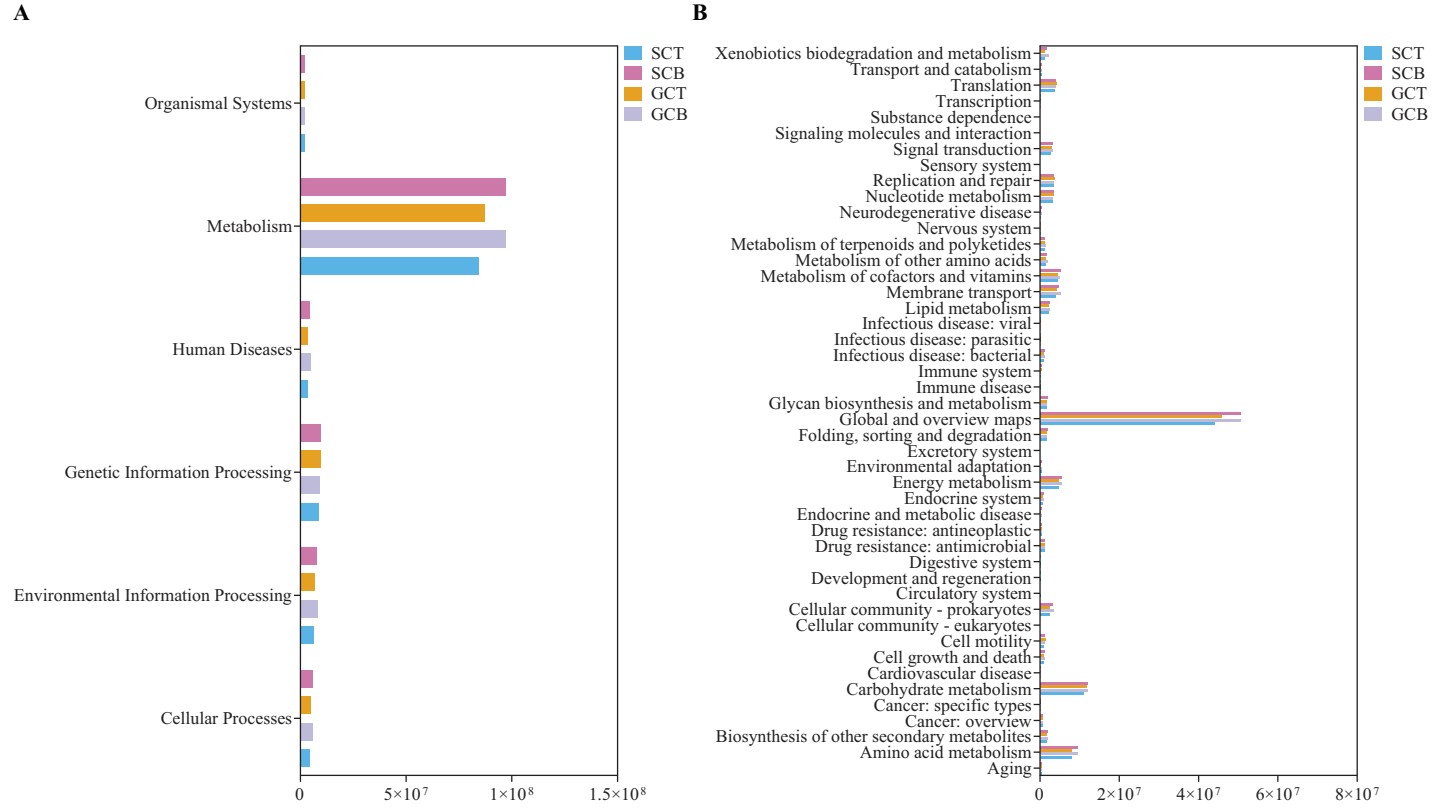

**Figure 6** PICRUSt prediction of KEGG metabolic pathway at Level 1 (A) and Level 2 (B) in four groups.

## DISCUSSION

In this study, we analyzed the fecal microorganisms of *L. nycthemera* and *C. pictus* living at Tianjin Zoo and Beijing Wildlife Park by 16S rDNA high-throughput sequencing technology. Analysis of fecal microorganisms facilitated our insight into the processes of nutrient utilization as well as the metabolic regulation of the hosts. In the comparison of alpha diversity and species abundance of gut bacteria, the differences between sample groups were not significant. In contrast, beta diversity analysis revealed significant differences in the structural composition of the gut microbiota between GCT and GCB.

### The effects of different environments (Beijing *vs* Tianjin) on the same species

The gut microbiota of *L. nycthemera* and *C. pictus* were mainly Firmicutes, which occupied the largest proportion in both Tianjin group and Beijing group. The result was consistent with reported studies on the gut microbiota of birds and mammals (*Hird et al., 2015*; *Oakley et al., 2014*). Firmicutes can digest proteins and break down complex carbohydrates, polysaccharides, and fatty acids, which facilitates the efficient absorption of energy and nutrients from food (*Clarke et al., 2014*). Our analysis revealed that the relative abundance of Firmicutes was significantly higher in GCT than that in GCB. Firmicutes can help *C. pictus* degrade fibers into volatile fatty acids, thus improving absorption capacity

(*Turnbaugh et al., 2009*). We analyzed dietary differences between living conditions, hoping to explain the difference on abundance of Firmicutes. The main food was pelleted feed and chopped vegetables in Tianjin Zoo, while the Beijing Wildlife Park mainly provides a wider variety of foods, including pheasant feed, seasonal vegetables, and fruits. Thus, golden pheasants had a broader source of energy and did not have a high demand for Firmicutes. The difference caused by diet was consistent with previous studies (*Bibbo et al., 2016*), for example, the change in the gut microbiota of great tits (*Parus major*) was induced by the dietary changes (*Davidson et al., 2020*).

Bacteroidota is often dominant in the mammalian gut microbiota and can degrade polysaccharides as well as polymers such as carbohydrates and plant cell walls (*Fujisaka, Watanabe & Tobe, 2023*; *Thomas et al., 2011*). In this study, the relative abundance of Bacteroidota was second only to Firmicutes in the composition of gut microbiota from both *L. nycthemera* and *C. pictus*, which is consistent with the previous study on gut microbiota of captive bharals (*Chi et al., 2019*). Studies on the gut microbiota of model mice have shown that a high ratio of Firmicutes/Bacteroidota improves the extraction efficiency of mice for food (*Clarke et al., 2012*; *Magne et al., 2020*). The Firmicutes/Bacteroidota ratios of *L. nycthemera* and *C. pictus* in the Tianjin Zoo were increased to adapt to the single food type for energy acquisition and allowed for suppression of intestinal pathogenic bacteria.

The abundance of Verrucomicrobia was significantly lower in SCT than that in SCB, and this phylum is mainly comprised of environmental microorganisms that are free-living and saccharolytic based on previous study (*Bergmann et al., 2011*). Thus, the difference in Verrucomicrobia phylum is primarily attributed to the varying living environments between Beijing Wildlife Park and Tianjin Zoo, including disparities ecological conditions.

The dominant genera varied among the four groups. GCT had a relatively high level of *Clostridium_sensu_stricto_1*, *Anaerosporobacter* and *Cellulosilyticum*. Among them, *Clostridium_sensu_stricto_1* is able to break down cellulose, while, the genus *Anaerosporobacter* was related with host heath. *Anaerosporobacter* belonging to family Lachnospiraceae, was found in the gut of broiler chickens previously, and could be used as a probiotic to enhance the efficacy of a vaccine against Campylobacter (*Nothaft et al., 2017*). *Cellulosilyticum*, could catabolize cellulose, is one of the probiotic species found in gut microbiota of hooded cranes (*Zhao et al., 2017*). In our study, the relative abundance of genus *Anaerosporobacter* and *Cellulosilyticum* from *C. pictus* were higher in Tianjin Zoo than that in Beijing Wildlife Park. The abundance of *Olsenella*, which could produce short-chain fatty acids to maintain the function of intestinal epithelial cells (*Wang et al., 2021*), is higher in GCB than in GCT. Therefore, even within the same species, there are variations in their capacity to regulate probiotics across different environments, consequently resulting in varying levels of animal health between different regions.

The abundance of *Bifidobacterium*, *Megamonas*, and *Solobacterium* in SCT were significantly higher than that in SCB. Various strains of *Bifidobacterium* which are the beneficial bacteria could use complex carbohydrates as the substrate. This genus has been reported to suppress diarrhea and could be utilized as probiotics (*Feng et al., 2019*).

In clinical trials, age, geographic origin, and race all directly influence the abundance of *Bifidobacterium* in the gut. The decrease of *Bifidobacterium* abundance is associated with a high intake of vegetables and diet (*Feng et al., 2019*), which may be the underlying cause for the differences observed between Tianjin and Beijing in our study. Members of *Megamonas* could produce acetic and propionic acid in rodents. It has been shown to be a substrate to form lipogenesis and cholesterol, which may affect weight loss rate in dogs (*Kieler et al., 2017*). Thus, in this study we speculated that different living environments provide different diets, leading to the diverse gut microbes of the same species. The role of these gut bacteria in *L. nycthemera* and *C. pictus* requires further experimental verification due to the absence of individual data such as weight and age.

### Differences in gut composition between species (*silver pheasants* and *golden pheasants*) under the same living environment

First, we compared the gut microbiota of *L. nycthemera* and *C. pictus* in Beijing Wildlife Park (SCB and GCB), and there were no significant differences in diversity or richness. The dominant flora in the two groups were Firmicutes followed by Bacteroidota in SCB and Actinobacteriota in GCB, respectively. Desulfobacterota was the most different phyla between SCB and GCB in Beijing Wildlife Park. *Jian et al. (2021)* revealed that the diet of laying hens supplemented with valine resulted in a significant reduction in the abundance of cecal pathogenic bacteria, such as *Deferribacterota*, improving intestinal health. Based on the preliminary analysis of gut microbiota, we found that the two species *L. nycthemera* and *C. pictus* differ in their ability to utilize cellulose and protein from food, but more data on amino acid metabolism are needed to detail the specific differences. At the genus level, the relative abundance of *Bifidobacterium* and *Subdoligranulum* in GCB were significantly higher than that in SCB. Based on previous studies, *Subdoligranulum*, one of the producers of butyrate, could protect the host health (*Chassard, de Wouters & Lacroix, 2014*). Many strains of Bifidobacterium also have been used to alter gut microbial ecology and improve host health. The *Bifidobacterium* has demonstrated important roles in the metabolism of host-derived glycans. Furthermore, probiotics have been reported to affect the gut-brain axis, as a common probiotic, *Bifidobacterium* may influence the functioning of the brain and central nervous system (*Presti et al., 2015*). Since the diet and living conditions were almost the same in this study, it is plausible that genetic variations between silver pheasants and golden pheasants at the Beijing Wildlife Park may underlie the observed differential abundance of *Bifidobacterium* and *Subdoligranulum*.

Next, we compared the gut microbiota of *L. nycthemera* and *C. pictus* in Tianjin Zoo (SCT and GCT), and the biggest difference in abundance of gut microbiota at the phylum level were Chloroflexi, Nitrospirota and Methylomirabilota. *Zhu et al. (2022)* found that the application of cotton straw biochar and Bacillus compound biofertilizer could improve the secretion of organic acids and amino acid compounds by Methylomirabilota and other strains. Another study showed that the relative abundance of Methylomirabilota increased in fully saturated soils, indicating the improvement of oxygenic denitrifiers, specifically NC10 members (*Schmitz et al., 2023*). Because Methylomirabilota can be commonly found in soil, *L. nycthemera* and *C. pictus* were under the same living conditions in Tianjin, the
different abundance of Methylomirabilota may be caused by the ability of soil microorganisms to colonize the intestinal tracts of the different species through the food consumed. At the genus level, the relative abundance of *Cellulosilyticum* in GCT was significantly higher than that in SCT. Previous studies revealed that *Cellulosilyticum* could convert cellulose into metabolites (*Zhao et al., 2017*). Our analysis revealed that avian gut microbial composition, which were fed the same diet and inhabited the same environment, was largely species dependent. This study demonstrates that the host is a dominant factor in shaping the microbial communities and the conclusion was similar with previous reports (*Fu et al., 2021*; *Garcia-Amado et al., 2018*). Importantly, this study provides basic research on the intestinal microflora of different avian species, which will be imperative for future studies.

## Metabolic analysis and functional prediction

These findings will help us understand the gut microbiota of *L. nycthemera* and *C. pictus*, and thus provide a theoretical basis for the protection and animal welfare. The differences in the metabolic pathways of *C. pictus* between Tianjin Zoo and Beijing Wildlife Park were significant ($P < 0.05$). In the prediction of synthetic function, the relative abundance of carbohydrate metabolism in GCB was higher than those in GCT, while, the relative abundance of amino acid metabolism pathways in GCB was lower than those in GCT. We hypothesized that the increased variety of food types in Beijing Wildlife Park led to a higher abundance of gut microbiota related to carbohydrate metabolism, such as *Christensenellaceae* related with the catabolism of cellulose and hemicellulose, and *Candidatus Saccharimonas* associated with cellulose degradation. These results suggested that the differences in diets could affect the components of the gut microbiota.

The results of our BugBase phenotypic prediction analysis showed that the relative abundance of Gram-negative bacteria in GCB was significantly higher than that in GCT, consistent with the high relative abundance of *Burkholderia-Caballeronia-Paraburkholderia* in GCB, which belongs to Proteobacteria. Some species in *Burkholderia-Caballeronia-Paraburkholderia* are pathogenic to humans and animals, by causing pulmonary infections and respiratory diseases (*Depoorter et al., 2016*). Moreover, the abundance of probiotics *Clostridium_sensu_ stricto_1* (*Doulidis et al., 2023*) and *Lactobacillus* (*Xiao et al., 2021*), were higher in GCT compared with GCB. Based on the analysis of the proportion of pathogenic bacteria and probiotics, we preliminary speculated the health of *C. pictus* under different living conditions. Combining the dominant bacteria analysis and phenotype prediction at the genus level, it is tentatively hypothesized that GCT improves digestion and absorption capacity, while GCB fecal microorganisms have a relatively higher abundance of pathogenic bacteria, which may lead to potential disease risk. Due to the presence of numerous unannotated gene sequences, which impede our comprehensive analysis of metabolic functions, the investigations of metabolomics and transcriptomics further will elucidate the association between fecal microorganisms and host species, including *L. nycthemera* and *C. pictus*.

## CONCLUSIONS

We analyzed the fecal microorganisms of *L. nycthemera* and *C. pictus* in Tianjin Zoo and Beijing wildlife Park using high-throughput sequencing. The main composition of the gut microbiota was consistent with the results of many bird studies, including Firmicutes and Bacteroidota. There were no significant differences in the diversity of the gut microbiota, but there were significant differences in the proportion of dominant bacteria at the genus level among four groups. These results suggested that the diets and living conditions affect the gut microbiota of these birds, as well as the functional differences in metabolism of host. The relative abundance of gut microbiota related to cellulose decomposition was higher in GCB than the GCT, indicating that the difference in cellulose content in the diet between Tianjin and Beijing is the major factor. The analysis on phenotypic prediction revealed that GCB has a potentially high risk of disease, which should attract zookeepers' attention on animal health. This study provides data for an in-depth understanding of the fecal microorganisms of silver pheasants and golden pheasants under different living conditions, and also provides a scientific reference for the use of gut microbiota to improve the health of captive animals.

## ACKNOWLEDGEMENTS

We appreciated the staff of Tianjin Zoo and Beijing Wildlife Park for their friendly support.

### Funding

This research was supported by the Tianjin Bureau of Planning and Natural Resources. The funders had no role in study design, data collection and analysis, decision to publish, or preparation of the manuscript.

### Grant Disclosures

The following grant information was disclosed by the authors:
Tianjin Bureau of Planning and Natural Resources.

### Competing Interests

The authors declare that they have no competing interests.

### Author Contributions

- Yushuo Zhang conceived and designed the experiments, performed the experiments, analyzed the data, prepared figures and/or tables, authored or reviewed drafts of the article, and approved the final draft.
- Xin He conceived and designed the experiments, performed the experiments, analyzed the data, authored or reviewed drafts of the article, and approved the final draft.
- Xiuhong Mo conceived and designed the experiments, performed the experiments, analyzed the data, authored or reviewed drafts of the article, and approved the final draft.

- Hong Wu conceived and designed the experiments, analyzed the data, prepared figures and/or tables, authored or reviewed drafts of the article, and approved the final draft.
- Dapeng Zhao conceived and designed the experiments, analyzed the data, prepared figures and/or tables, authored or reviewed drafts of the article, and approved the final draft.

## Data Availability

The data are available at the NCBI Sequence Read Archive (SRA): PRJNA941118.

## Supplemental Information

Supplemental information for this article can be found online at http://dx.doi.org/10.7717/peerj.16979#supplemental-information.

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
