# Peer review of "Similarities and differences: species and diet impact gut microbiota of captive pheasants"

_PeerJ, doi:10.7717/peerj.16979_

## Round 0.1 · original submission · Major Revisions

Both reviewers have identified a number of issues with the manuscript, in the methods and results, that require addressing before this manuscript can be accepted. Some of these focus around the amount of detail provided, but there were also some concerns raised around the statistics that need to be addressed.

Reviewer 1 ·

Basic reporting

Overall, the paper titled “Similarities and Differences: Species and Diet Impact Gut Microbiota on Endangered Pheasants” addresses an intriguing aspect of gut microbiota of L. nycthemera and C. pictus in two different zoos. The research topic is interesting, and the writing is good. However, there are many areas that require attention and need to be addressed before the paper can be considered for publication.

Experimental design

1. The authors used a Student’s t-test for comparisons between different groups and considered p<0.05 statistically significant. It’s advised to use Wilcoxon rank sum test instead of t-test given that microbiome data is not normal distributed. Moreover, the p value for multiple comparison should be corrected and consider FDR<0.05 as statistically significant.

2. It’s unclear how the authors conducted the metabolic analysis and functional prediction given that they only have 16s rRNA sequence data in this study. The authors should be more careful about their wording and provide details for functional prediction.

Validity of the findings

1. The access to raw data of 16s rRNA sequence is not available.

2. The quality of the main figures needs to be improved. The images appear to be low-resolution, resulting in unclear visibility of many data points.

Additional comments

42-43: It should be non-invasive sampling instead of “non-injury sampling”.

86-87: Bacteroides and Alistipes should be written in italic.

144-146: Details are needed for the microbiome data quality control and profiling. Information such as quality score and microbial database used for taxa identification should be mentioned.

177-178: “There were significant differences between the four groups from different species/living conditions.” This statement doesn’t make any sense here and should be provided with more information and citations for readers to understand.

224-225: It's confusing as why alpha diversity index was calculated based on the Student’s t-test. The authors should explain this.

226-228: “Under the same living conditions, the ACE and Chao indices were both higher in SCT (824.4 and 839, respectively) compared with GCT (659.9 and 633, respectively), while the Simpson index was lower in SCT (0.06) compared with GCT (0.13).” Are the indices significantly different between groups?

235-236: How were the coverage indices calculated? The authors should provide more information to explain it.

242-243: “The results showed that there was a significant difference between GCT and GCB (R =0.4222, P = 0.005), leading the complete separation between these two groups.” The statistics for the weighted PcoA and unweighted PcoA are different. The authors should report the information for specific figures.

243-245: “Moreover, there was no significant difference in beta diversity among individuals of L. nycthemera from different living conditions (P > 0.05, Supplementary Figure 6).” The word “however” is more appropriate than “moreover” in this content.

248-251: “Based on the 16S rDNA sequencing results, the functional composition of COG was relatively similar in all samples” How to get the conclusion? No figures or tables were provided for the statement.

Figure 6 and Figure 7: PICRUSt prediction using 16s sequence is not accurate at all. All these annotation figures should be removed or moved to supplementary Figures.

Reviewer 2 ·

Basic reporting

General comments:
This paper focuses on a pilot survey of the composition and diversity of the fecal microbiota in two pheasant species. The research is original, and the research questions are well defined. The methods in this research are sufficient to answer the research questions. However, English writing should be improved significantly. I strongly suggest that authors invite a fluent English speaker to proofread their manuscript.
Based on the current manuscript quality, this manuscript should be accepted with major revisions.
Specific Comments:
Abstract:
L40: “In recent years”. Consider using “In recent years,” (missing a comma).
L44: “composition and diversity of the fecal microbiota”. Consider revising to “bacterial composition and diversity of the …..”. Based on this manuscript, the scope of this research is within the Domain of Bacteria. I suggest that the authors go over this manuscript to emphasize this.
L45-47: “The results showed that ……. phylum level.” This sentence needs to be rephrased. It is not concise and clear.
Introduction:
Generally, the goal of the study needs to be more clearly defined. There is sufficient background information about this research and the background information is well connected to the current study. The introduction could add more background knowledge related to the two pheasant species’ diets and living environments.
L78-79: “under the same environment”. The author should give details of these environmental factors.
L81: “surroundings”. It is not clear what “surroundings” means here.
L110-114: This sentence needs to be revised and make the research goal and scope clearly defined.
Materials and Methods:
Generally, more information about sampling, sequencing, and bioinformatics is needed. Well-organized writing and more related citations are needed.
The raw sequencing reads need to be deposited to NCBI SRA and an access number should be provided. Furthermore, the authors should include all R scripts for analyses including Venn diagrams, taxonomy plots (e.g., Fig 2), t-tests, and bar plots (e.g., Fig 3, 4, etc.) as a part of supplemental materials.
L121: “non-invasive sampling technique”. The authors need to give detailed information or citations of this non-invasive sampling technique.
L125: “international and national guidelines”. If the authors used any international and national guidelines, please cite these guidelines here.
L135-136: Again, the authors should clearly outline the scope of this research. It is not clear what these two primers target. Do they target Bacteria, Archaea, or Fungi?
L137: “ddH2O”. Revise to “ddH2O”. I recommend the authors go over the whole manuscript and check the formats.
L139: “Illumina miseq”. Revise to “Illumina MiSeq”.
L144: “L. nycthemera and C. pictus”. Species names should be in italics. I recommend the authors go over the whole manuscript and check the formats and spelling.
L145: “using QIIME (version 1.9.1)”. Software QIIME citation is missing here. Please cite:
J Gregory Caporaso, Justin Kuczynski, Jesse Stombaugh, Kyle Bittinger, Frederic D Bushman, Elizabeth K Costello, Noah Fierer, Antonio Gonzalez Pena, Julia K Goodrich, Jeffrey I Gordon, Gavin A Huttley, Scott T Kelley, Dan Knights, Jeremy E Koenig, Ruth E Ley, Catherine A Lozupone, Daniel McDonald, Brian D Muegge, Meg Pirrung, Jens Reeder, Joel R Sevinsky, Peter J Turnbaugh, William A Walters, Jeremy Widmann, Tanya Yatsunenko, Jesse Zaneveld and Rob Knight; Nature Methods, 2010
L144-147: “The microbial communities …. high-throughput sequencing (D'Amore et al., 2016).”
A lot of information is missing here. For example, it is unclear whether the authors used Illumina MiSeq paired-end or single-ended sequencing. How did authors do quality control (i.e., QC criteria) on the raw reads? Which database did authors use for assigning taxonomy? The authors only mentioned RDP for classification, but they did not mention the databases for assigning taxonomy. The authors used QIIME to do the quality control, assign taxonomy and build the OTU table. Ideally, the QIIME scripts used need to be mentioned in the Methods section. I suggest the authors look up and cite the following three papers (check the “Methods” sections for details), and adopt a clear approach to describe QIIME workflow.

Jiang et al. (2022) Limits to the three domains of life: lessons from community assembly along an Antarctic salinity gradient
Jiang, X., and Takacs-Vesbach, C.D. (2017) Microbial community analysis of pH 4 thermal springs in Yellowstone National Park. Extremophiles 21: 135-152.
Van Horn, D.J., Wolf, C.R., Colman, D.R et al. (2016) Patterns of bacterial biodiversity in the glacial meltwater streams of the McMurdo Dry Valleys, Antarctica. FEMS Microbiol Ecol 92.
L151-158: Here also lacks detailed information about alpha and beta diversity analyses. Normally, alpha (e.g., Chao1, ACE, Shannon, etc.) and beta (e.g., UniFrac) diversity analyses require dataset normalization. This can be easily done by rarefying to equal depth (i.e., an equal number of sequences across samples). Based on the writing (Line 158), the authors have normalized OTU tables already, but they need to mention this here too. Please refer to the above three papers and see how to address the normalization.
L151-152: Missing Mothur software citation. Please cite the paper below:
Schloss, P. D., Westcott, S. L., Ryabin, T., Hall, J. R., Hartmann, M., Hollister, E. B., … others. (2009). Introducing mothur: open-source, platform-independent, community-supported software for describing and comparing microbial communities. Appl. Environ. Microbiol., 75(23), 7537–7541.
L159: Missing PICURSt software citation. Please cite this paper:
Langille, Morgan G I; Zaneveld, Jesse; Caporaso, J Gregory; McDonald, Daniel; Knights, Dan; Reyes, Joshua A; Clemente, Jose C; Burkepile, Deron E; Vega Thurber, Rebecca L; Knight, Rob; Beiko, Robert G; Huttenhower, Curtis (2013). Predictive functional profiling of microbial communities using 16S rRNA marker gene sequences, Nature Biotechnology. 31 (9): 814–821
L160-161: “the database of orthologous groups (COG) and Kyoto encyclopedia of genes and genomes (KEGG)”. Please check the full names of the two databases. The names are incorrect. For example, COG stands for “Cluster of Orthologous Groups of proteins”.
L160-161: Missing the citations of COG and KEGG databases. Please cite these papers:
Tatusov RL, Galperin MY, Natale DA, Koonin EV (2000) The COG database: a tool for genome-scale analysis of protein functions and evolution. Nucleic Acids Res 28:33-36
Kanehisa, M. and Goto, S.; KEGG: Kyoto Encyclopedia of Genes and Genomes. Nucleic Acids Res. 28, 27-30 (2000).
Kanehisa, M; Toward understanding the origin and evolution of cellular organisms. Protein Sci. 28, 1947-1951 (2019)
Kanehisa, M., Furumichi, M., Sato, Y., Kawashima, M. and Ishiguro-Watanabe, M.; KEGG for taxonomy-based analysis of pathways and genomes. Nucleic Acids Res. 51, D587-D592 (2023)
L162: “… phenotypic prediction using BugBase (Langille et al., 2013).” The citation of BugBase software is not correct. Please cite this paper:
Ward T, Larson J, Meulemans J, Hillmann B, Lynch J, Sidiropoulos D, Spear JR, Caporaso G, Blekhman R, Knight R, Fink R, Knights D (2017) BugBase predicts organism-level microbiome phenotypes. bioRxiv:133462
Results:
The authors wrote detailed research results. The results are solid and statistical analyses are robust.
L166-167: “A total of 24 fecal samples from both and 1,700,701 optimized sequences were obtained …..”.
The authors mentioned the number of sequences and the average length after QIIME quality control. The authors should also report basic sequencing results and qualities such as the number of raw reads, the average length, etc. before the filtering and quality control. The authors should include this information.
Discussion:
Some suggestions on the discussion:
Authors need to add more discussion to make their discussion more robust. As a pilot survey, what is the significance of the current results? Can we apply the research results to future pheasant conservation? Authors discovered Gram + and Gram – bacteria were significantly different in different groups. A further discussion/explanation is needed for this.
L270-271: “alpha diversity and species abundance, the differences between the four groups of samples were not significant.” It is not clear whether diversity and species abundance mean bacterial diversity/abundance.
L285-290: The authors pointed out that the food in zoos is different from the food in natural environments. Still, it is not clear what food wild pheasants eat. How can different food cause different microbial compositions?
L375-377: “These results suggest that…”. This sentence is not clear. It needs to be rephrased.
L378-379: “The results of our BugBase phenotypic prediction analysis showed that the relative ……..higher than in GCT”.
The authors mentioned that Gram + and Gram – bacteria were significantly different among different groups in the Results and Discussion sections. Gram bacteria is an old concept, whereas the authors used molecular sequencing methods in this research. Each group (Gram + or Gram -) has many phyla. It is ambiguous to use Gram bacteria in this manuscript and readers cannot get much information. I advise the authors to remove all the “Gram bacteria” terms in this manuscript and focus on the different diversity/abundance in phyla level, class level, etc.
Tables and Figures:
Figures 3, 4, and 6: The fonts on the X and Y – axes are too small.
Supplementary Materials:
All supplemental tables and figures do not have captions. Clear descriptive captions are needed.
The raw sequencing reads need to be deposited to NCBI SRA and an access number should be provided.
The authors should include all R scripts for analyses including Venn diagrams, taxonomy plots (e.g., Fig 2), t-tests, box plots, and bar plots (e.g., Fig 3, 4, etc.) as a part of supplemental materials.

Experimental design

The research is original, and the research questions are well defined. The methods in this research are sufficient to answer the research questions.

Validity of the findings

The results are solid and statistical analyses are robust.

Annotated reviews are not available for download in order to protect the identity of reviewers who chose to remain anonymous.

---

## Round 0.2 · Major Revisions

While the authors have made a decent attempt at addressing most of the reviewers' comments, I disagree with your argument regarding the use of t-test vs. Wilcoxon rank test. You acknowledge in the rebuttal letter that Student's t-test is sensitive to normality and distribution remains normal as long as the sample size is large enough. It is highly unlikely that your data is normally distributed, and the sample size is only n=6. While you have included analysis by the Wilcoxon test, the subsequent paragraphs, detailing which taxa are significantly higher/lower in each sample are very difficult to follow and do not address how these results compare to your t-test results. I am not going to ask reviewers to wade through this and compare themselves with your initial results in the manuscript. If you insist on using the t-test, please provide in the rebuttal document an easy-to-follow table of each significant result by t-test, and whether this changes or remains significant by Wilcoxon test (plus any taxa that become significant by Wilcoxon that were not by t-test).

A couple of additional comments:
1. The formula that you included for coverage indices. While I don't feel that you don't need to include the formula in the manuscript, could you include a reference to this in the manuscript?
2. "248-251: “Based on the 16S rDNA sequencing results, the functional composition of COG was relatively similar in all samples” How to get the conclusion? No figures or tables were provided for the statement."

You say this was visualised using boxplots. So how was significance determined?

Please just edit the current rebuttal letter for the next revision, for reviewers to be able to see all changes.

---

## Round 0.3 · Major Revisions

While the authors have made a significant effort to improve the quality of the manuscript, both reviewers still have significant issues with the statistical analyses of the data, including a lack of multiple test correction or rarefaction of the data.

Reviewer 2 mentions the lack of variance and maximum/minimum number of reads per sample. Is this supposed to be what Table S1 is? If so, I have some additional concerns about this table. You have Seq_num as a column header - is this the number of paired-end reads? Is there a row missing (SCT1)? You give a mean length of reads - many of these are over 400bp, but the Illumina MiSeq produces a maximum read length of 300bp.

Were your sequencing libraries spiked with PhiX? How were these reads removed? Was a positive control (mock microbial community) used, to check primers didn't preferentially amplify some genera over others? Was a negative control used (starting with the DNA extraction) to check for contamination?

Reviewer 1 ·

Basic reporting

The paper titled 'Similarities and Differences: Species and Diet Impact Gut Microbiota in Endangered Pheasants' addressed most of the reviewer's comments, but there are still a few areas that require attention before it can be accepted for publication.

Experimental design

1. Details for microbiome data processing, such as quality control and profiling, should be provided in the manuscript.
2. Authors are advised to seek help from a fluent English speaker or professional service to proofread their manuscript, as it still contains grammar errors.

Validity of the findings

1. The authors accepted the suggestion to use the Wilcoxon rank-sum test, but they didn't report whether the p-value for multiple comparisons has been corrected. If not, the authors should correct the p-values for multiple comparisons and consider FDR < 0.05 as statistically significant.

Additional comments

NA

Reviewer 2 ·

Basic reporting

Dear Editor,
The manuscript has undergone revisions, yet certain critical issues concerning bioinformatics and statistics highlighted in the initial review have not been adequately addressed. These issues possess the potential for significant statistical biases.
Therefore, I strongly recommend a further revision of the manuscript before considering it for publication.
In the 2nd edition of the manuscript:
L187-188: “A total of 24 fecal samples from both species were analyzed and 1,700,701 optimized sequences were obtained,”
L160-170: “Alpha diversity indices including Chao, Ace, Shannon and Simpson indices were calculated using Mothur (version 1.30.2) to reflect the abundance and diversity of microbial communities (Schloss et al., 2009). The coverage indices calculated using Mothur reflect whether the sequencing results represent the real situation of the microorganisms in each sample”
In the 1st edition of the manuscript:
L144-147: “The microbial communities …. high-throughput sequencing (D'Amore et al., 2016).”
The authors have not effectively addressed these statistical issues (specifically related to sample sizes) in the two editions of the manuscript. The authors only mentioned the total number of reads generated for 24 samples, but they didn’t mention the variance of the number of reads among samples. What is the maximum number of reads of the sample? What is the minimum number of reads of the sample? The Illumina MiSeq platform does not generate an equal number of reads across samples. However, the authors used “the real situation of the microorganisms in each sample” for alpha diversity indices, which means without rarifying to the equal number of reads across samples. Moreover, the authors did not mention whether reads were rarified for beta diversity index calculations.
The variance of sample sizes (the number of reads) will result in serious statistical biases and can distort the whole picture of the results. I recommend the authors refer to and cite the three papers mentioned in the first review feedback (refer to the detailed review comments on L144-147 in the first edition of the manuscript, which mentioned, “A lot of information is missing here. For example, ….Van Horn, D.J., Wolf, C.R., Colman, D.R et al. (2016)'").
Particularly, the authors must utilize a rarified subsample dataset, ensuring an equal depth (i.e., the same number of reads) before conducting any statistical analyses, such as alpha and beta index calculations, and PICRUSt analysis. Without rarefying samples, there is a risk of low statistical power, inflated false discovery rates, inflated effect size estimations, and low reproducibility.

Experimental design

NA

Validity of the findings

NA

Additional comments

NA

Annotated reviews are not available for download in order to protect the identity of reviewers who chose to remain anonymous.

---

## Round 0.4 · Minor Revisions

An improvement on the previous version, but a couple of remaining minor changes to make.

1. Adjust the methods to state have you corrected for multiple corrections (and if P-values in figures 3 and 4 are based on FDR as well, this should be stated in these legends also).

2. I can see you have included a statement regarding public release of data (line 139), so that's fine.

3. Table S1: are you missing "sequencing reads" from the table legend here?

4: Table 2 legend: Add in taxa - Mean relative abundance of the 10 most abundant taxa at the phylum level (same for table 3).

5. The title you have given Table S1 in our system (The total number of raw reads, base pairs, the mean length of the reads) does not match what is in that table. In your rebuttal letter, you mentioned the amplicon was sequenced from both ends - so this is the length of the amplicon, not the length of the read).

6. Rebuttal letter answers questions about the controls and phiX, but you have not made this clear in the manuscript. Also, this is not transcriptome

Reviewer 1 ·

Basic reporting

The manuscript has been enhanced through revision, and the authors have addressed the majority of comments from reviewers. However, several areas still need to be addressed before it can be accepted for publication.

Experimental design

1. The authors indicated that the p-value for multiple comparisons has been corrected. They are supposed to include this information in their methods section.

Validity of the findings

1.Raw reads should be deposited publicly, and a data availability statement should be provided in the manuscript.

Reviewer 2 ·

Basic reporting

Dear Editor,

The manuscript has been significantly improved after the revision. The authors have addressed all the questions and concerns. Therefore, I recommend that this manuscript should be formally accepted and published.

Experimental design

NA

Validity of the findings

NA

Additional comments

NA

---

## Round 0.5 · accepted · Accept

Thank you for addressing the remaining concerns in this manuscript. It is now ready for publication.